# Severe Intrahepatic Cholestasis of Pregnancy—Potential Mechanism by Which Fetuses Are Protected from the Hazardous Effect of Bile Acids

**DOI:** 10.3390/jcm12020616

**Published:** 2023-01-12

**Authors:** Gal Hershkovitz, Yael Raz, Ilana Goldinger, Ariel Many, Liran Hiersch, Rimon Eli

**Affiliations:** 1Department of Obstetrics and Gynecology, Lis Hospital for Women’s Health, Tel Aviv Sourasky Medical Center, Affiliated to the Sackler Faculty of Medicine, Tel Aviv University, Tel Aviv 6423906, Israel; 2Department of Clinical Biochemistry Laboratory, Tel Aviv Sourasky Medical Center, Affiliated to the Sackler Faculty of Medicine, Tel Aviv University, Tel Aviv 6423906, Israel; 3Department of Obstetrics and Gynecology, Mayanei Hayeshua Medical Center, Bnei Bark, Israel, Affiliated to the Sackler Faculty of Medicine, Tel Aviv University, Tel Aviv 51544, Israel

**Keywords:** intrahepatic cholestasis of pregnancy, total bile acid, ursodeoxycholic acid, placenta

## Abstract

Intrahepatic cholestasis of pregnancy (ICP) is characterized by elevated total bile acids (TBA). Although elevated maternal TBA is a major risk factors for fetal morbidity and mortality, it is unclear why some fetuses are more prone to the hazardous effect of bile acids (BA) over the others. It is unclear whether fetuses are protected by placental BA uptake, or it is the fetal BA metabolism that reduces fetal BA as compared to maternal levels. Therefore, we aimed to compared TBA levels in the umbilical vein and artery to maternal TBA in women with ICP. The study included 18 women who had TBA > 40 μmol/L and their 23 fetuses. We found that the TBA level in umbilical vein was significantly lower compared to maternal TBA level. The TBA levels in umbilical vein and umbilical artery were similar. No fetus had a serious neonatal complication. Importantly, since TBA level remains low even though maternal TBA level is high the fetuses are protected from the hazardous effects of maternal BA. Our findings suggest that there is no effective metabolism of BA in the fetus and the main decrease in TBA in the fetus is related to placental BA uptake.

## 1. Introduction

Intrahepatic cholestasis of pregnancy (ICP) usually occurs at the third trimester and is characterized by pruritus, elevated liver enzymes, and elevated total bile acids (TBA) >10 µmol/L and resolves soon after birth [1,2]. The etiology of ICP is multifactorial but a crucial cause is gene mutations in the hepatocellular transporters that are responsible for the transfer of bile acids (BA) to the bile canaliculi which result in accumulation of BA in maternal hepatocytes and consequently in maternal blood [3,4]. Among the genetic factors the most common are mutation in ABCB4 (MDR3) (16% of all ICP cases), ABCB-11 transport pump (BSEP), ABCB4 protein, ATP8B 1 and TJP2 genes [1,3]

Several studies have suggested that maternal excessive BA level leads to both maternal and fetal complications in correlation to maternal TBA level [5,6,7]. ICP was shown to be associated with increased risk for spontaneous preterm labor, meconium-stained amniotic fluid (MSAF), non-reassuring fetal heart rate, fetal asphyxia and perinatal deaths [5,6]. Importantly, fetuses who are exposed to maternal TBA > 100 µmol/L are in the greatest risk to develop fetal complications mainly stillbirth [8,9], which is a major concern and the main reason for indicated preterm delivery in those cases. Fetal death is thought to be directly related to bile acid effects on the fetal heart as well as on the chorionic vessels [1]. Interestingly, although maternal TBA level > 40 µmol/L (and even more so > 100 µmol/L) are a major risk factors for fetal compromise, most of the fetuses who are exposed to high TBA levels are born without major complications [8,9]. Therefore, it was suggested that these fetuses are protected from the toxic effect of excessive BA by either uptake of BA in the placenta, BA metabolism in fetal liver or the therapeutic effect of UDCA [1,10]. The underling mechanism of this protective effect is unclear, and data are limited. Beouwer et al. [7] have shown that TBA level in umbilical vein was lower compared to maternal TBA level supporting the role of the placenta in the homeostasis of bile acids. In addition, it was reported that during intrauterine life fetal hepatobiliary excretory function is not efficient while the main route for the elimination of BA is their transfer to the mother across the placenta [10]. Yet, previous studies were limited mainly by low sample size [7,11,12], e lack of information on maternal treatment [7], investigation of umbilical vein but not both umbilical vein and artery for the assessment of fetal TBA levels [7,12] and the lack of data on neonatal outcome [11,12].

In the current prospective study, we aimed to compare TBA levels in the umbilical vein and artery to maternal TBA levels in women with severe ICP in order to investigate the potential role of the placenta in the protection of fetuses from high TBA levels. In addition, we aimed to compare the TBA level in umbilical vein to TBA level in umbilical artery. This comparison may explore the role of BA metabolism in fetal compartment as related data are quite limited [6].

## 2. Materials and Methods

This prospective study was conducted between 2020–2022 at the Tel Aviv Sourasky Medical Center, which is a tertiary referral center with >11,000 annual deliveries. The incidence of ICP in our population is estimated as 0.3% [13].

### 2.1. Study Design

#### Diagnosis and Management of Women with ICP

According to our local protocol, pregnant women with elevated liver enzymes and/or pruritus are tested routinely for serum TBA levels. In addition, a thorough investigation is undertaken to rule out preeclampsia, TORCH infection (Toxoplasmosis, Rubella, Cytomegalovirus, Parvo virus, Herpes virus, Syphilis), viral hepatitis (hepatitis B, C), and other liver or biliary disease. The diagnosis of ICP is based on pruritus, elevated liver enzymes, and elevated TBA levels (>10 µmol/L) in the absence of other possible causes [14,15]. The severity of ICP was defined according to TBA levels: mild (10–20 µmol/L), moderate (20–40 µmol/L) or severe (>40 µmol/L) [1,5]. All women who were diagnosed with severe ICP at <37 weeks of gestation were hospitalized in our Maternal-Fetal Medicine Department for maternal and fetal monitoring. This includes frequent fetal non-stress and/or biophysical profile and twice weekly blood test for liver enzyme and TBA levels. All women with ICP received UDCA (Ursolit; CTS Chemical Industries Ltd., Kiriat Malachi, Israel) usually at initial dose of 300–600 mg × 3 per day and the dose of UDCA was raised up to a maximal dose of 900 mg × 3 per a day. If the response was still inadequate, a second line of therapy was added using Rifampin or Cholestyramine. All women who were diagnosed with severe ICP before 34 weeks have received corticosteroids for fetal lung maturation. In addition, our common practice is to deliver at 37 weeks of gestation all women with ICP, in line with the widely accepted guidelines [1,14,15]. Women with severe ICP who did not respond to medical treatment were delivered at 35–37 weeks [1,14]. If TBA level increased to >100 µmol/L despite treatment, delivery was considered before 34 weeks in selected cases.

### 2.2. Study Population

Since severe ICP (TBA >40 μmol/L) is a major risk factor for fetal complications we focused on women with TBA level >40 μmol/L within 36 h prior to delivery documented after a night fasting. We excluded women for whom TBA level within 36 h before delivery was not available.

Eligible women were approached and recruited for the study following obtaining an informed consent. After delivery of the fetus, blood samples were collected from the umbilical vein and artery within 5 min of cord clamping.

### 2.3. Specimen Collection and Handling

Blood samples were collected into vacutainer tubes containing clot activator. The tubes were centrifuged for 10 min at 2000× *g* at 25 degrees Celsius. The serum was stored at 4 degrees Celsius for a few hours until analysis performed. The cord blood cord samples were collected into vacutainer tubes containing clot activator and tubes were centrifuged for 10 min at 2000× *g* at 25 degrees Celsius. The serum was stored at 4 degrees Celsius for no more than 24 h until analysis performed.

### 2.4. TBA Assay

The quantitative determination of TBA was performed on the ADVIA 2400 Clinical Chemistry System (Siemens Healthcare, Erlangen, Germany) with the use of the Diazyme Laboratory Total Bile Acids Assay Kit (DZ042A-K; Diazene Lab, Poway, CA, USA) by 2 experienced technicians.

### 2.5. Data

Maternal and neonatal data were obtained from the computerized database. Maternal data included: age, parity, body mass index (BMI), obstetric and medical history, gestational age at diagnosis, gestational age at first administration of UDCA, TBA at diagnosis, maximal TBA level, TBA level within 24–36 h before delivery, gestational age at delivery, and mode of delivery. Neonatal data included: birthweight, birthweight percentile and the rate of small for gestational age birthweight < 10 percentile) (SGA) (calculated using a local growth chart) [16], Apgar score at 5 min. Fetal complications which are suggested to be associated with ICP were also documented including evidence for MSAF, fetal asphyxia (pH < 7 and base deficit >12 mmol/L), spontaneous preterm delivery and perinatal death. We also collected data on the rate of transient tachypnea of the neonate (TTN), respiratory distress syndrome (RDS, admission to neonatal intensive care unit (NICU) and TBA levels in the umbilical vein and umbilical artery.

### 2.6. Statistical Analysis

Statistical analysis and graphic presentation were performed with the use of the GraphPad Prism software (version 9; GraphPad, San Diego, CA, USA) and IBM SPSS statistical software (version 27; IBM, New York City, NY, USA). For baseline description of TBA values in maternal and fetal blood, median and interquartile range (IQR) were calculated.

Fisher’s exact test and Mann-Whitney test were used for comparison of non-parametric variables. Binary Binary logistic regression models were also performed. Two-sided tests were used and a probability value of <0.05 was considered statistically significant for all tests performed. The correlation between TBA levels in maternal and fetal blood was calculated using the Spearman’s correlation coefficient and the fit of a non-linear regression line.

### 2.7. Ethical Approval

The study was approved by our local Institutional Review Board protocol number (090/2020).

## 3. Results

Overall, 18 women and 23 fetuses were included in the study. Maternal characteristics and outcome are summarized in Table 1. The mean gestational age at the diagnosis of ICP was 32 ± 5 weeks and mean TBA at diagnosis was 33.4 ± 3 µmol/L. Most women (88%) delivered <37 weeks of gestation and 3 (17%) delivered <34 weeks. Five women delivered by CS: 3/5 of women with twin pregnancy (1-non elective and 2 elective) and 2/13 of women with singleton (elective CS). Three women delivered spontaneously (1 at 35 weeks and 2 at 37 weeks) and among 15 women labor was induced due to poor response to medical treatment. As for UCDA treatment, 3 women received 600 mg × 3, 7 received 900 mg and 6 got a second line therapy. Mean maternal TBA at delivery was 76.4 ± 30.3 µmol/L and in 22% of women the levels exceeded 90 µmol/L. The fetal TBA levels and neonatal outcome is summarized in Table 2. The mean TBA levels in the umbilical vein were significantly lower compared to maternal TBA level at delivery (22.5 ± 8.8 vs. 76 ± 30 μmol/L, respectively, *p* = 0.01). The mean feto-matrnal gradient between maternal and fetal TBA levels was 67 ± 8.6% and this phenomenon was consistent even among women with extremely high TBA level > 90 µmol/L (Figure 1). Separate umbilical vein and umbilical artery samples were available for 19 out of 23 fetuses. There was no significant difference between mean TBA levels in umbilical vein and artery (22.5 ± 8.8 vs. 19.5 ± 6.5 µmol/L, respectively, *p* = 0.09). Out of 23 neonates 7 (30%) had MSAF and 6 (26%) were admitted to the NICU (3 due to TTN and 3 due to mild RDS). There was no case of perinatal death or fetal asphyxia. There was no difference in neonatal parameters at the time of delivery including median gestational week at delivery, median birth weight percentile, maternal BA, Umbilical vein BA, percent of BA decrease between maternal blood and UV and meconium between singleton and twin neonates (Appendix A). Using a binary logistic regression model, there were no variables predictive of meconium (Appendix A). Appling binary logistic regression models, we did not find any predicting factors for meconium. We investigated the correlation between the maternal and fetal TBA levels. There was a positive correlation between TBA level in umbilical artery and umbilical vein (Figure 2A, r = 0.78, *p* < 0.0001). We also found a positive correlation between umbilical vein TBA level and maternal TBA level at delivery (Figure 2B, r = 0.52, *p* = 0.01).

## 4. Discussion

In this prospective study we have shown that among women with severe ICP at delivery the TBA levels in the umbilical vein was significantly lower as compared to maternal serum TBA. In addition, there was no significant difference between TBA levels in umbilical vein and umbilical artery.

Only a few small sample studies investigated the correlation between maternal TBA level and cord blood TBA levels, usually without comparing the levels of umbilical vein and artery. Mazzela et al. [12] showed that among 15 untreated women cord blood TBA levels was lower compared to those of maternal serum. In line with our results the umbilical TBA was lower compared to maternal TBA in group of women treated with UDCA. Our results are further supported by a recent study by Brouwers et al. [7] which reported that the mean TBA level among 24 women with ICP with TBA > 40 µmol/L). The mean umbilical TBA was lower compared to mean maternal TBA level for moderate (40–99 µmol/L) and severe ICP (>100 µmol/L) but the authors did not specifically compare the maternal TBA to their fetuses TBA levels. In addition, the authors did not compare level in umbilical vein and artery. The mean TBA level in the umbilical vein of fetuses in cases of maternal moderate and severe ICP were 9 and 14 µmol/L, respectively, which are similar to the umbilical TBA levels in our cohort. In contrast, Geenes et al. [11] reported that in women with ICP who were treated with UDCA fetal TBA levels were not significantly lower than maternal levels. However, the data were based on a small cohort of only 5 women, and mean maternal TBA was less than 10 μmol/L. It should be noted that we included in our study only women with severe ICP for whom the TBA level at delivery was available. This group of women is in high risk for fetal complications and, hence, it is of most importance to investigate the bile acid metabolism in the feto-maternal system. We carefully measured maternal TBA levels as close as possible to delivery (no more than 36 h before delivery) and, therefore, this TBA value represented the BA concentration to which the fetus was exposed during delivery. We also provided an important data on fetal complications including known ICP associated complications, i.e.: spontaneous preterm delivery, MSAF and fetal asphyxia. There was no difference in maternal and neonatal outcome among singleton and twin pregnancies. An additional analysis has revealed that there were no variables predictive of meconium.

Importantly, we managed to have umbilical vein samples for 23 fetuses and both umbilical vein and artery samples for 19 fetuses out of these 23 fetuses. This is a higher number of participants compared to similar studies published earlier [6,7,12]. Interestingly, we found no significant difference between TBA levels in umbilical vein and umbilical artery. Geenes et al. [6] also reported no significant difference between TBA levels in umbilical vein and artery but only in a small number of women with ICP (5–7 women). Therefore, our finding is highly important, and it suggests that fetal liver metabolism does not play a crucial role in the metabolism of bile acids being transferred from maternal origin. These findings are further supported by studies which reported that during intrauterine life fetal hepatobiliary excretory function is not efficient while the main route for the elimination of BA is their transfer to the mother across the placenta [1,10,17,18].

In addition to placental favorable effect on metabolism of BA documented in untreated women it was suggested that UDCA treatment may have an additional role in the homeostasis of BA in feto-maternal unit [1,10,19]. Noteworthy, several studies have already shown that UDCA can reduce the incidence of feta complication among women with ICP [20] Indeed, UDCA was found to enhance the feto- maternal excretion of through a direct effect on BA active transporters in the placenta [21,22] and reduces maternal TBA level [1,6,22]. Therefore, UDCA can minimize the exposure of fetuses to high concentration of BA by activation of BA transporters in maternal hepatocytes and encourage the placenta to transfer BA back to maternal compartment.

Taken together the data suggest that placental excretion of bile acid during maternal cholestasis plays a major role in the protection of the fetus against the toxic bile acids existing in the maternal side while fetal BA metabolism is of minor importance. This mechanism can be enhanced by UDCA. This has an important implication, because even in situations of high maternal TBA there is only a mild increase in bile acid concentrations in fetal serum. In 15 of 18 women labor was induced due to poor response to medical treatment: 3 women received 600 mg × 3, 7 received 900 mg and 6 got a second line. These data indicate that these women had a severe active disease. However, despite their severe disease the TBA in umbilical vein and artery were still low compared to maternal TBA level. In our opinion this finding further supports the role of the protective effect of the placenta.

We assume that among women who have experienced perinatal mortality the fetuses are exposed to high maternal TBA level but due to unknown reasons such as placental abnormal excretory function or resistance to UDCA the fetal TBA remains high leading to fetal complications. Obviously, since we did not have any case of severe fetal complications or a case of IUFD during the period when the study was conducted this hypothesis waits for further reassurance.

### Strength and Limitations

This is a prospective study which includes only women with severe ICP for whom TBA level at delivery were available, a reliable estimation of maternal TBA level during delivery. This is the group of women who are in a risk for fetal complications and therefore, it is important to understand the influence of maternal TBA level on fetuses. In addition, the study provided data on separate samples from umbilical vein and artery for 19 women, a number of women that is higher than reported in previous studies. Therefore, the sample size still allowed us to investigate the role of fetal BA metabolism in homeostasis of BA in the feto-maternal unit. In contrast to previous studies, the current study provide data on fetal complications which are considered to be associated with ICP. The main limitation of study is the number of fetuses included in the study that is relatively small but higher than reported in previous studies. Nevertheless, since the difference between maternal TBA and fetal TBA were so prominent we were able to draw conclusions. In our institution all women were treated with UDCA and hence it remains unclear whether the low fetal TBA is attributed only to original protective effect of the placental excretory function or additional effect of UCDA. The fact that no fetus had a severe complication or IUFD makes it difficult to draw a conclusion regarding the correlation between fetal TBA level and major fetal complications. It should be noted that we did not find a study that investigated the fetal association between fetal TBA level and fetal complications. We believe that as we continue the study and recruit more women, we will be able to address this question. Finally, it should be noted that other placental mediated mechanisms for fetal complications, unrelated to BA uptake, were suggested including downregulation of iNOS expression in placenta which may results in poor utero-placental vasodilatation and fetal hypoxia [23].In this study we did not investigate the BA profile in maternal and cord blood samples. Geens et al. [11] have suggested that the BA profile in fetal serum is different from their mothers but the meaning of this finding is not clear. We did not investigate the BA profile in maternal and fetal serum. However, it should be noted that in a recent Cochrane Database publishment the authors stated that they have not found any compelling evidence to recommend or refute the routine use of bile acids profile in clinical practice since there were too few studies to permit a precise estimate of the accuracy of serum bile acid profile components [24]. We believe that the importance of BA profile in pathogenesis of ICP and the association between changes in BA profile in fetal serum and fetal complication require further investigation.

## 5. Conclusions

In the current prospective study, we focused on women with severe ICP whom offspring are in a significant risk for various complications. We have shown that among women with severe ICP TBA levels in both umbilical vein and artery were significantly lower compared to TBA level at delivery. In addition, we have demonstrated that there are no significant differences between the TBA levels in umbilical cord artery and vein. This finding suggests that there is no significant effective fetal liver BA metabolism to cope with the high flow of BA being transferred from maternal origin. Therefore, it is suggested that the level of BA in the fetal compartment remains relatively low even though maternal TBA level is high due to protective effect of the placenta. The mechanism by which the placenta protects the fetus from elevated maternal TBA level most probably involves active uptake of bile acids by the placenta mediated by specific BA transporters. Obviously, we will have to recruit more women and fetuses to support our hypothesis. Importantly, increased sample size will enable us to investigate the correlation between fetal TBA level and fetal complications. A thorough placental analysis in future cases of severe fatal complications might reveal the reason for placental dysfunction.

## Figures and Tables

**Figure 1 jcm-12-00616-f001:**
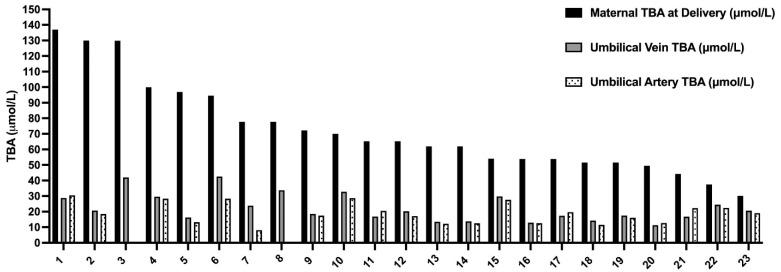
The total bile acids levels in maternal serum delivery, umbilical vein, and umbilical artery. The TBA level in maternal serum at delivery (black) compared to TBA levels in umbilical vein (gray, 23 samples) and umbilical artery (dotted, 19 samples). TBA—total bile acids in μmol/L.

**Figure 2 jcm-12-00616-f002:**
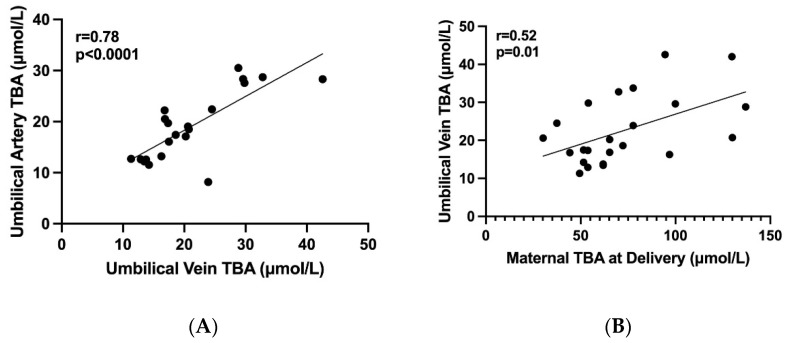
The correlations between maternal and fetal TBA levels. Graphic diagrams which show the correlations between maternal and fetal TBA levels. (**A**) A positive correlation between TBA levels in umbilical artery and vein (r = 0.78, *p* < 0.0001), (**B**) A positive correlation between TBA levels in maternal serum at delivery and umbilical vein (r = 0.52, *p* = 0.01), TBA—total bile acids in μmol/L.

**Table 1 jcm-12-00616-t001:** The maternal demographics and outcome.

Age (years) at Time of Delivery ^a^	5.2 ± 36.5
Parity ^a^	0.9 ± 1
Pregestational BMI ^a^	3 ± 23
Multiple gestation ^b^	(27) 18/5
ICP in previous pregnancy ^b^	(44) 18/8
IUFD in previous pregnancy due to ICP ^b^	(5.5) 18/1
Gestational diabetes ^b^	6/18 (33)
Gestational age at diagnosis (wks.) ^a^	5 ± 32
TBA at diagnosis ^a^ (µmol/L) (range)	(94−12) 3 ± 33.4
Gestational age at UDCA administration	33 ± 5.7
Length of treatment (days) ^a^	29 ± 44 (2−133)
Maximal TBA ^a^ (µmol/L) (range)	99.6 ± 46 (51−247)
Second line therapy	6/18(33)
Maternal TBA at delivery ^a^ (µmol/L) (range)	76.4 ± 30.3 (40−129)
Maternal TBA at delivery >90 µmol/L. ^b^	4/18(22)
Gestational age at delivery (wks.) ^a^	35.6 ± 1.6
Delivery < 37 wks. ^b^	16/18 (88)
Delivery < 34 wks. ^b^	3/18 (17)
Spontaneous delivery ^b^	1/18 (5.5)
Cesarean section ^b^	5/18 (27)

^a^ Data are presented as mean ±SD. ^b^ Data are presented as n (%). BMI—Basic Metabolic Index, IUFD-intrauterine fetal demise, ICP-intrahepatic cholestasis of pregnancy, UDCA-Ursodeoxycholic acid, TBA—Total bile acids, UDCA ursodeoxycolic acid.

**Table 2 jcm-12-00616-t002:** The neonatal outcome.

Umbilical Vein TBA ^a^ (µmol/L), (range)	22.5 ± 8.8.(12−42)
Umbilical artery TBA ^a^ (µmol/L), (range)	19.6 ± 6.5 (11−30)
Birthweight singleton (gr) ^a^	2614 ± 380
Birthweight percentile singleton ^a^	51 ± 22
Birthweight twins (gr) ^a^	2440 ± 466
Birthweight percentile twins ^a^	57 ± 26
SGA ^b^	0/23
Umbilical cord pH <7.1 ^b^	0/23
MSAF	7/23 (30)
Fetal asphyxia ^b^	0/23
NICU admission ^b^	6/23 (26)
NICU hospitalization (days) ^a^, (range)	24 ± 11 (3−36)
RDS ^b^ (gestational age at delivery, wks.)	3/23 (13) (32−33)
Jaundice ^b^	5/23 (21)
Perinatal death ^b^	0/23

^a^ Data are presented as n(%) or mean (±SD) where appropriate. ^b^ Data are presented as n (%). TBA—total bile acid, SGA—Small for Gestational Age. MSAF—meconium-stained amniotic fluid, NICU—Neonatal Intensive Care. RDS—respiratory distress syndrome.

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
