# Peer review of "Severe Intrahepatic Cholestasis of Pregnancy—Potential Mechanism by Which Fetuses Are Protected from the Hazardous Effect of Bile Acids"

_jcm, 2023, doi:10.3390/jcm12020616_

Round 1
Reviewer 1 Report
General comments:
The paper entitled “Severe intrahepatic cholestasis of pregnancy - potential mechanism by which fetuses are protected from the hazardous effect of bile acids” has been carefully reviewed. The value of the current study could have been increased if a control group has been also included. The paper is rather well written and structured, although there are a lot of typing errors and/or mistakes. Nonetheless, the paper has a merit of providing important information in the field. It compares the bile acids levels in fetal umbilical vein and artery to the levels in the maternal serum. Considering the title of the manuscript, unsatisfying that the authors did not discuss or highligh neither in the core of the manuscript nor in the conclusion the potential mechanism by which fetuses are protected from the hazardous effects of bile acid in ICP. The authors have neglected discussing all aspects related to transplacental transport of the bile acids (BAs). Based on these aspects, improvements are required prior to the acceptance for the paper for publication.
Specific comments
1. The authors concluded that “our findings suggest that there is no significant metabolism of BA in the fetus and the main decrease in Total BAs (TBA) in the fetus is related to the placental metabolism.” Did the authors compare the profile of TBA in the fetal artery versus fetal vein, and these versus maternal TBA?
2. It is established that the fetal liver synthesizes BAs from weeks 14 to 16 of pregnancy onwards. In the light of this knowledge, how and on which evidences the authors explain the conclusion drawn in the current study suggesting the lack of significant metabolism of BA in the fetus?
3. It would be beneficial for the readers’ understanding if the authors extend the introduction section of the current study. The authors should highlight if BAs are intrinsically hazardous molecules and/or indicate if the hazardous effect of BAs discussed (i.e. in ICP) are specific to extremely elevated circulating levels of BAs during the studied disease (ICP)? These details are very important for the readers. Indeed, BAs are described in several studies since 1999 onwards as ligands targeting nuclear receptors (e.g. FXR, see Parks et al. 1999, in Science., 284, pp. 1365-1368, ‘Bile acids: natural ligands for an orphan nuclear receptor’) and cell-surface receptors (e.g. TGR5 see Kawamata et al. 2003, In J. Biol. Chem., 278, pp. 9435-9440, ‘A G protein-coupled receptor responsive to bile acids’).
4. The authors should pay more attention to details. They should avoid diverse typing errors, unnecessary space between words and commas, dots, etc., present throughout the manuscript (ex., see lines 23, 45, 49, 68, 180, 260, 287, 295, etc).
5. The authors must be systematic in using descriptive statistics. This is not yet the case: see line 180 (p=01) and line 185 (p=0.09).
6. Table 1:
a) The data mean ± SD (e.g. Age and Parity) should be presented the same way with or without after commas.
b) Some of the data in brackets are not explained in the footnote (e.g. under TBA at diagnosis; length of treatment
7. Discussion:
Line 238: the authors should indicate at which days of term the TBA was measured. The current statement “as close as possible to delivery” is not specific enough. Indeed, specific values are required to help a reader to better understand the implications of the study but also to make comparisons between data described here with other data. This precision would also help to reproduce the study.
Lines 244-246 and 248-250: it is important that the authors extend the discussion of their findings on the light of a well-known evidence that fetal liver synthesizes BA from weeks 16 to 17 onwards, and that the profile of BA in maternal serum is different from the bile acid profile in fetal umbilical cord (see Geenes et al. 2014, PlosOne 9 (1)e83828), respectively.
8. Conclusions
Line 297-298: did the authors make sure that the TBA profile in umbilical artery and umbilical vein is unchanged?
Author Response
General comments:
The paper entitled “Severe intrahepatic cholestasis of pregnancy - potential mechanism by which fetuses are protected from the hazardous effect of bile acids” has been carefully reviewed. The value of the current study could have been increased if a control group has been also included. The paper is rather well written and structured, although there are a lot of typing errors and/or mistakes. Nonetheless, the paper has a merit of providing important information in the field. It compares the bile acids levels in fetal umbilical vein and artery to the levels in the maternal serum. Considering the title of the manuscript, unsatisfying that the authors did not discuss or highligh neither in the core of the manuscript nor in the conclusion the potential mechanism by which fetuses are protected from the hazardous effects of bile acid in ICP. The authors have neglected discussing all aspects related to transplacental transport of the bile acids (BAs). Based on these aspects, improvements are required prior to the acceptance for the paper for publication.
Specific comments
- The authors concluded that “our findings suggest that there is no significant metabolism of BA in the fetus and the main decrease in Total BAs (TBA) in the fetus is related to the placental metabolism.” Did the authors compare the profile of TBA in the fetal artery versus fetal vein, and these versus maternal TBA? We did not analyze the profile of TBA in the mother and the fetus.
- It is established that the fetal liver synthesizes BAs from weeks 14 to 16 of pregnancy onwards. In the light of this knowledge, how and on which evidences the authors explain the conclusion drawn in the current study suggesting the lack of significant metabolism of BA in the fetus? We are not saying that there is no BA metabolism in the fetus, but it is premature and not effective . We changed our statements in lines 29 and 349 to effective instead of 'no active" We already discussed it in line 61-63 and 276-281.
- It would be beneficial for the readers’ understanding if the authors extend the introduction section of the current study. The authors should highlight if BAs are intrinsically hazardous molecules and/or indicate if the hazardous effect of BAs discussed (i.e. in ICP) are specific to extremely elevated circulating levels of BAs during the studied disease (ICP)? These details are very important for the readers. Indeed, BAs are described in several studies since 1999 onwards as ligands targeting nuclear receptors (e.g. FXR, see Parks et al. 1999, in Science., 284, pp. 1365-1368, ‘Bile acids: natural ligands for an orphan nuclear receptor’) and cell-surface receptors (e.g. TGR5 see Kawamata et al. 2003, In J. Biol. Chem., 278, pp. 9435-9440, ‘A G protein-coupled receptor responsive to bile acids’). It is well established that fetal complications are strongly associated with TBA level. It is described in the introduction. However, the mechanism by which BA leads to fetal complication is not clear. Anyway, it is not within the focus of our paper.
- The authors should pay more attention to details. They should avoid diverse typing errors, unnecessary space between words and commas, dots, etc., present throughout the manuscript (ex., see lines 23, 45, 49, 68, 180, 260, 287, 295, etc). We corrected any mistake presented by the reviewers.
- The authors must be systematic in using descriptive statistics. This is not yet the case: see line 180 (p=01) and line 185 (p=0.09). In line 194 we had a mistake p=0.01 and mot p=01
- Table 1:
- a) The data mean ± SD (e.g. Age and Parity) should be presented the same way with or without after commas. Corrected
- b) Some of the data in brackets are not explained in the footnote (e.g. under TBA at diagnosis; length of treatment- We added an explanation for UDCA, but we believe everything is quite clear in table 1.
- Discussion:
Line 238: the authors should indicate at which days of term the TBA was measured. The current statement “as close as possible to delivery” is not specific enough. Indeed, specific values are required to help a reader to better understand the implications of the study but also to make comparisons between data described here with other data. This precision would also help to reproduce the study. Please note in lines 103-104 , we clearly stated " Since severe ICP (TBA >40 μmol/L) is a major risk factor for fetal complications we focused on women with TBA level >40 μmol/L within 36 hours prior to delivery..." . Indeed, this is one of the strengths of the study. We added in line 262 " no more than 36 hours before delivery".
Lines 244-246 and 248-250: it is important that the authors extend the discussion of their findings on the light of a well-known evidence that fetal liver synthesizes BA from weeks 16 to 17 onwards, and that the profile of BA in maternal serum is different from the bile acid profile in fetal umbilical cord (see Geenes et al. 2014, PlosOne 9 (1)e83828), respectively.
We discuss this issue in lines 333-342 and we also provide a Cochrane review (ref 24 ) which does not support the use of BA profile in the diagnosis of ICP.
- Conclusions
Line 297-298: did the authors make sure that the TBA profile in umbilical artery and umbilical vein is unchanged? We did not analyze the profile of TBA in the mother and the fetus. Although the importance of different BA profile in the fetus and women is not clear we revised the "discussion" section section to clarify our massage.
Reviewer 2 Report
Comments for the authors
In this paper, the authors investigated the potential mechanism by which fetuses are protected from the hazardous effect of bile acids in severe intrahepatic cholestasis of pregnancy.
I have some recommendations to look for regarding this article:
- More than 50% of references are too old.
- Line 31 typo mistake.
- Line 35 - should be entered together with the third trimester, the appearance of ICP in the second trimester, and the remission at birth.
- Line 43 preterm labor instead of premature labor. It often uses the terminology preterm labor, premature birth.
Discussion
o Nitric oxide synthase iNOS plays an important role in poor fetoplacental vascular perfusion and adverse pregnancy outcomes. iNOS can provide complementary information in predicting the extent and severity of ICP. DOI: 10.12659/MSM.930176
o Increased maternal TBA levels in PE/HELLP pregnancies indicate a relation between bile acids in the maternal circulation and HELLP syndrome. As overall TBA levels in maternal blood are increased compared to UCB, we conclude that the placenta partly protects the fetus from increased maternal TBA levels. This consistent difference in TBA levels between the maternal and fetal compartments is unrelated to the placental expression of ABCG2. DOI: 10.1016/j.bbadis.2014.11.008
- Under normal metabolic conditions, there are strong associations between bile acid profiles and glucose homeostasis. doi: 10.1111/1471-0528.15926.
- Being a prospective study, why is the control group missing??
- Specify what the exclusion and inclusion criteria were
- Possible exclusion criteria - other causes of liver dysfunction (hemolysis, HELLP syndrome, preeclampsia, acute fatty liver of pregnancy, acute/chronic viral hepatitis, primary biliary cirrhosis, biliary obstruction on ultrasound).
- A limitation of the study is given by the lack of data regarding the mode of sample collection (fasting or postprandial) since TBA measured by enzyme assay can increase 2-5 times, reaching a peak approximately 90 minutes after the table.
- Table 1 gestational diabetes 6/23 (33); do you mean probable 6/18?
- What was the mode of delivery for the other 12 patients?
- No incidence mentioned 0.2-2% with seasonal pattern
- Line 163, please define the groups/subgroups
- How many cases were classified as mild, moderate, and severe forms?
- Only 5 cesarean sections in the whole lot under the conditions in which there were also 5 twin pregnancies?
o Twin pregnancies with ICP were associated with a higher rate of cesarean section and preterm birth than those without ICP or singleton pregnancies.
o Wu et al. showed a cesarean section rate of 56.14% in a singleton pregnancy and 93.67% in a twin pregnancy. doi: 10.1080/07853890.2022.2136400
- Labor induction should be recommended from 37+0 weeks or 34+0 to 36+6 weeks when the bile acid concentration is above >100 µmol/l (expert consensus; consensus strength +++).
- Were there any maternal and neonatal outcomes differences between singleton and twin pregnancies?
- If it was a prospective study, why don't we also have pre-pregnancy BMI because it has been proven that there is a correlation with the incidence of ICP
- What was the HBsAg positive status?
- In Figure 2 (A) you state that "A positive correlation between TBA levels in umbilical artery and vein (r=0.78, p<0.0001)," and in lines 244-246, you stated that "we did not find no significant difference between the umbilical vein and umbilical artery TBA levels."
- Line 257 typo mistake feta instead of fetal
- In the abstract, you stated that "Our findings suggest that there is no significant metabolism of BA in the fetus, and the main decrease in TBA in the fetus is related to placental metabolism." "We hypothesize that among women who have experienced perinatal mortality, fetuses are exposed to a high level of maternal TBA, but for unknown reasons, such as abnormal placental excretory function or UDCA resistance, fetal TBA remains high, leading to fetal complications. …as we had no cases of severe fetal complications or IUFD during the study period, this hypothesis awaits further reassurance.” The conclusions of a study must be clear and concise.
Kindest regards

Author Response
- More than 50% of references are too old.
The "old references" (Ref 1,3,4 17,23,26) were deleted or replaced by newer studies as suggested by the reviewer. We reconsidered the importance of other " old references" (5,13,19,20,24 ) and we decided to keep them due to their valuable massage.
- Line 31 typo mistake.- corrected.
- Line 35 - should be entered together with the third trimester, the appearance of ICP in the second trimester, and the remission at birth.-corrected
- Line 43 preterm labor instead of premature labor.-Corrected .
Discussion
o Nitric oxide synthase iNOS plays an important role in poor fetoplacental vascular perfusion and adverse pregnancy outcomes. iNOS can provide complementary information in predicting the extent and severity of ICP. DOI: 10.12659/MSM.930176
o Increased maternal TBA levels in PE/HELLP pregnancies indicate a relation between bile acids in the maternal circulation and HELLP syndrome. As overall TBA levels in maternal blood are increased compared to UCB, we conclude that the placenta partly protects the fetus from increased maternal TBA levels. This consistent difference in TBA levels between the maternal and fetal compartments is unrelated to the placental expression of ABCG2. DOI: 10.1016/j.bbadis.2014.11.008
- Under normal metabolic conditions, there are strong associations between bile acid profiles and glucose homeostasis. doi: 10.1111/1471-0528.15926.
We added a paper about iNOS - ref 23 and in the text lines 327-330.
- Being a prospective study, why is the control group missing??
We planned to compare cases of severe complications to cases with no sever complications but finally we did not have severe complications .In addition, the number of women with TBA level >100 was small and did not enable a sub analysis. This point is reported as one of the limitations of the study.
- Specify what the exclusion and inclusion criteria were
- Possible exclusion criteria - other causes of liver dysfunction (hemolysis, HELLP syndrome, preeclampsia, acute fatty liver of pregnancy, acute/chronic viral hepatitis, primary biliary cirrhosis, biliary obstruction on ultrasound).
All women had ICP. We did not have women with other liver disease or HELLP/preeclampsia. The diagnosis protocol is described in the methods.
- A limitation of the study is given by the lack of data regarding the mode of sample collection (fasting or postprandial) since TBA measured by enzyme assay can increase 2-5 times, reaching a peak approximately 90 minutes after the table.
All samples were taken following fasting , lines 105-106.
- Table 1 gestational diabetes 6/23 (33); do you mean probable 6/18? Corrected.
- What was the mode of delivery for the other 12 patients?
Five women delivered by CS : 3/5 of women with twin pregnancy (1-non elective and 2 elective) and 2/13 of women with singleton ( elective CS) . Three women delivered spontaneously ( 1 at 35 weeks and 2 at 37 weeks) and among 15 women labor was induced due to poor response to medical treatment. As for UCDA treatment, 3 women received 600 mgX3 , 7 received 900 mg and 6 got a second line therapy, Lines 185-190 .
No incidence mentioned 0.2-2% with seasonal pattern.- We don’t have this data .
- Line 163, please define the groups/subgroups-
We reviewed the "statistical analysis" section lines 167-177.
- How many cases were classified as mild, moderate, and severe forms?
All women recruited to the study had TBA > 40 which was defined as severe ICP.
- Only 5 cesarean sections in the whole lot under the conditions in which there were also 5 twin pregnancies?
Women were basically healthy and except for ICP did not have other pregnancy complications which requires CS. The CS rate in our institution is 22 % for singleton and 72% for twins which are similar to data reported in our study. We added in lines 184-185.
o Twin pregnancies with ICP were associated with a higher rate of cesarean section and preterm birth than those without ICP or singleton pregnancies.
o Wu et al. showed a cesarean section rate of 56.14% in a singleton pregnancy and 93.67% in a twin pregnancy. doi: 10.1080/07853890.2022.2136400
- Labor induction should be recommended from 37+0 weeks or 34+0 to 36+6 weeks when the bile acid concentration is above >100 µmol/l (expert consensus; consensus strength +++).
Delivery before 36 weeks is in agreement with the statement of ACOG and RCOG:
"Delivery between 34 and 36 weeks of gestation can be considered in women with ICP, with total bile acid levels of 100 mmol/L, and with any of the following: excruciating and unremitting maternal pruritus not relieved with pharmacotherapy; a history of stillbirth before 36 weeks of gestation due to ICP with recurring ICP in the current pregnancy; or preexisting or acute hepatic disease with clinical or laboratory evidence of worsening hepatic function. " (ACOG).
Indeed, delivery before 34 weeks is not a formal consensus but rather a local policy. Therefore, we put the ACOG and RCOG references before the sentence regarding delivery before 34 weeks line 97-101.
- Were there any maternal and neonatal outcomes differences between singleton and twin pregnancies?
There was no difference in maternal or neonatal outcome but since numbers of cases were relatively small the reliability of the results is limited and therefore, we did not include the data in the original manuscript. Now we added a sentence in the results line 201-206.
.- If it was a prospective study, why don't we also have pre-pregnancy BMI because it has been proven that there is a correlation with the incidence of ICP
It is already included in table 1 under BMI. We corrected to pregestational BMI in table 1.
- What was the HBsAg positive status? There was no case of positive HBV.
- In Figure 2 (A) you state that "A positive correlation between TBA levels in umbilical artery and vein (r=0.78, p<0.0001)," and in lines 244-246, you stated that "we did not find no significant difference between the umbilical vein and umbilical artery TBA levels."
These 2 statements are not contradictory. Indeed, there was no significant difference between the umbilical vein and umbilical artery TBA levels. However, if TBA levels in umbilical vein was higher average so the TBA level in umbilical artery was also higher but still not statistically different .
- Line 257 typo mistake feta instead of fetal- The term is feto-maternal.
- In the abstract, you stated that "Our findings suggest that there is no significant metabolism of BA in the fetus, and the main decrease in TBA in the fetus is related to placental metabolism." "We hypothesize that among women who have experienced perinatal mortality, fetuses are exposed to a high level of maternal TBA, but for unknown reasons, such as abnormal placental excretory function or UDCA resistance, fetal TBA remains high, leading to fetal complications. …as we had no cases of severe fetal complications or IUFD during the study period, this hypothesis awaits further reassurance.” The conclusions of a study must be clear and concise.
This is only a hypothesis and as we stated it needs further support. This is not a conclusion of the study.
Reviewer 3 Report
1. Abstract: line 18“characterized by elevated total bile acids” ICP is not only characterized by this is one of the outcomes of disease.
2. Line 29: The “Placental metabolism” of BA is not confirmed by experiments.
3. Introduction: line 38: mention which transporter gene mutations are important and related to ICP.
4. Line 68-70: Must be changed, this is not correct. You are not hypothesizing this, actually, you are reconfirming the previous findings with more no. samples, material treatment information and neonatal outcomes.
5. Material methods: line 89 “per dayand .” should be corrected
6. My other serious concern about the methods are author did not mention, at what stage and on what treatment they analyzed patients for TBA levels. How many were respondents and how many were non-respondents, how did they handle them for statistics with such a sample no. of samples, which patients were at which level of disease, treatment and outcome of new-born?
7. If only the level of TBA is study, then what is new in this study than the previous one as the author mentioned in ref. 7,12,13.
8. Apparently, the author has all data in database, authors can rewrite or reanalysed or report this.
9. Results: Line184: “TBA levels in umbilical vein and artery (22.5±8.8 vs. 19.5± 6.5 µmol/L, respectively, p=0.09).” It looks like there will be a significant difference, the no. of samples needs to increase.
10. Line 212: Can the authors make similar correlations with the outcome of the neonates?
11. Line 219: Discussion is missing for disease state, treatment and neonate outcomes.
12. Line 235: how the author categorized the server disease. Only TBA levels are not sufficient to conclude the severity of the disease. Other patients may be on treatment. The author should either correct this statement and his interpretations
13. Author is suggesting maternal excretion is a major mechanism, and suggests UDCA may involve in this. In this case, then analysis of the BA composition is very important. I strongly suggest the authors analyze the BA composition and measure the concentration of UDCA. Then, you can predict in discussion line 263.
14. Conclusions are farfetched. For example which complication authors discuss which was associated with the ICP. Then, the author is talking about the liver metabolism of BA just with BA levels in the blood. I strongly suggest that author must rewrite the conclusion according to the findings.
Over all comment
What is more novel in this study than if it already reported that TBA level in the umbilical vein was lower compared to maternal TBA level as the author gave ref. also Beouwer et al. [7]. Sample no. in this ref. is comparable with this study. So, the author is reconfirming the previous finding and suggesting he has more no. of sample and material treatment information and data on neonatal outcomes. This is partially incorrect, I suggest changing is interpretation and conclusions more critically and correctly. The author should discuss this more in this paper as a strength of his study, especially in the abstract & other sections etc.
Author Response
- Abstract: line 18“characterized by elevated total bile acids” ICP is not only characterized by this is one of the outcomes of disease.
Indeed, pruritus occurs due to elevated TBA. Our statement is referring to the presentation of disease and not to cause and results.
- Line 29: The “Placental metabolism” of BA is not confirmed by experiments.
We accepted the reviewer remark and we changed to placental BA uptake instead of metabolism in lines: 21,30,56.
- Introduction: line 38: mention which transporter gene mutations are important and related to ICP.
We added in line 41-43
- Line 68-70: Must be changed, this is not correct. You are not hypothesizing this, actually, you are reconfirming the previous findings with more no. samples, material treatment information and neonatal outcomes.
We accept the reviewer remark and we decided to delete the sentence lines 69-71
- Material methods: line 89 “per dayand .” should be corrected- Corrected.
- My other serious concern about the methods are author did not mention, at what stage and on what treatment they analysed patients for TBA levels. How many were respondents and how many were non-respondents, how did they handle them for statistics with such a sample no. of samples, which patients were at which level of disease, treatment and outcome of new-born?
We added data to the results about treatment and response lines 185-190.
- If only the level of TBA is study, then what is new in this study than the previous one as the author mentioned in ref. 7,12,13.
We revised the section (lines 242-267 which explains the specific design of our study compared to the three other studies to clarify our massage.
- Apparently, the author has all data in database, authors can rewrite or reanalysed or report this.???
- Results: Line184: “TBA levels in umbilical vein and artery (22.5±8.8 vs. 19.5± 6.5 µmol/L, respectively, p=0.09).” It looks like there will be a significant difference, the no. of samples needs to increase.
Indeed, the issue of number of women included in the study is one of the limitations of the study and we discussed it in the discussion.
- Line 212: Can the authors make similar correlations with the outcome of the neonates? We added the results of an analysis lines 201-207 and in discussion .264-267.
- Line 219: Discussion is missing for disease state, treatment and neonate outcomes.
The data on neonatal outcome is already reported in Table 2.We added data on 185-190 on disease state and treatment" in the results . we added in the discussion a sentence line 295-300.
- Line 235: how the author categorized the server disease. Only TBA levels are not sufficient to conclude the severity of the disease. Other patients may be on treatment. The author should either correct this statement and his interpretations.
All women included had pruritus ( we replaced clinical presentation in line 82 by itching to clarify) and elevated liver enzymes . We defined the severity of ICP as commonly suggested and we added the specific references in line 85.
- Author is suggesting maternal excretion is a major mechanism, and suggests UDCA may involve in this. In this case, then analysis of the BA composition is very important. I strongly suggest the authors analyse the BA composition and measure the concentration of UDCA. Then, you can predict in discussion line 263.
Indeed, the data on the composition of BA in maternal and fetal blood might be important but we do not have the data on the composition of the bile acids.
- Conclusions are farfetched. For example which complication authors discuss which was associated with the ICP. Then, the author is talking about the liver metabolism of BA just with BA levels in the blood. I strongly suggest that author must rewrite the conclusion according to the findings.
We revised the conclusions, and we believe that the conclusion is supported by the findings.
We added additional analysis of factors which affect fetal complications which are associated with ICP.
Over all comment
What is more novel in this study than if it already reported that TBA level in the umbilical vein was lower compared to maternal TBA level as the author gave ref. also Beouwer et al. [7]. Sample no. in this ref. is comparable with this study. So, the author is reconfirming the previous finding and suggesting he has more no. of sample and material treatment information and data on neonatal outcomes. This is partially incorrect, I suggest changing is interpretation and conclusions more critically and correctly. The author should discuss this more in this paper as a strength of his study, especially in the abstract & other sections etc.
We revised the "discussion" section section to clarify our massage.
Round 2
Reviewer 2 Report
Line 188 how was labor induced?
Line 194 typo mistake – feto-maternal
Line 196 what happened to the other four fetuses?
Line 227 excludes – figure 1, and on line 233 – figure 2, enter each figure's legend.Author Response
1.line 188 – the protocol of labor induction is described in lines 101-102
line 178 feto-maternal- it is an accepted phrase.
LINE 196- for 4 fetuses umbilical artery sample were not available due to technical failure.
Line 227 – Corrected in line 229 and 235
Reviewer 3 Report
Line 201-207: Please show this data in supplementary fills, it will be very useful for the reader.
Author Response
Table were added for the supplement tables 3+4 and lines 203-210 were revised.